# SARS-CoV-2 Production, Purification Methods and UV Inactivation for Proteomics and Structural Studies

**DOI:** 10.3390/v14091989

**Published:** 2022-09-08

**Authors:** Zlatka Plavec, Aušra Domanska, Xiaonan Liu, Pia Laine, Lars Paulin, Markku Varjosalo, Petri Auvinen, Sharon G. Wolf, Maria Anastasina, Sarah J. Butcher

**Affiliations:** 1Molecular and Integrative Biosciences Research Programme, Faculty of Biological and Environmental Sciences, University of Helsinki, 00790 Helsinki, Finland; 2Institute of Biotechnology, Helsinki Institute of Life Science, University of Helsinki, 00790 Helsinki, Finland; 3Department of Chemical Research Support, Weizmann Institute of Science, Rehovot 76100, Israel

**Keywords:** SARS-CoV-2, purification, size exclusion, UV inactivation, structural studies

## Abstract

Severe acute respiratory syndrome coronavirus-2 is the causative agent of COVID-19. During the pandemic of 2019–2022, at least 500 million have been infected and over 6.3 million people have died from COVID-19. The virus is pleomorphic, and due to its pathogenicity is often handled in very restrictive biosafety containments laboratories. We developed two effective and rapid purification methods followed by UV inactivation that allow easy downstream handling of the virus. We monitored the purification through titering, sequencing, mass spectrometry and electron cryogenic microscopy. Although pelleting through a sucrose cushion, followed by gentle resuspension overnight gave the best particle recovery, infectivity decreased, and the purity was significantly worse than if using the size exclusion resin Capto Core. Capto Core can be used in batch mode, and was seven times faster than the pelleting method, obviating the need for ultracentrifugation in the containment laboratory, but resulting in a dilute virus. UV inactivation was readily optimized to allow handling of the inactivated samples under standard operating conditions. When containment laboratory space is limited, we recommend the use of Capto Core for purification and UV for inactivation as a simple, rapid workflow prior, for instance, to electron cryogenic microscopy or cell activation experiments.

## 1. Introduction

Severe acute respiratory syndrome coronavirus 2 (SARS-CoV-2) is the cause of the coronavirus disease (COVID-19) pandemic, which started at the end of 2019 and is still ongoing in 2022. Despite prior infections and/or vaccinations, the fraction of the susceptible population is high, further ensuring the evolution of SARS-CoV-2 and prolonging the pandemic. Thus, continuing research is required to support further vaccine development and drug discovery [1].

Generating a purified sample of concentrated, intact virions is often a necessity for virology research. Production of viral stocks for downstream applications relies on suitable cell culture models. In the case of SARS-CoV-2, these are epithelial cell lines originating from humans or monkeys, such as Calu-1, Calu-3, Caco-2, and Vero cell line derivatives or genetically modified A549 cells expressing virus receptors [2,3,4,5]. These cell lines support SARS-CoV-2 production by secreting progeny virions into the culture medium. However, they may also secrete mediators of the antiviral response, such as interferons and/or proinflammatory cytokines [5,6,7]. When crude virus stocks are used for infection experiments, these paracrine signaling molecules can trigger a multitude of pathways, resulting in high background noise and affecting the experimental readout and data interpretation, especially when highly sensitive ‘omics approaches are used [8,9,10]. Biophysical and structural biology experiments such as cryogenic electron microscopy (cryoEM) and single-particle reconstruction, also typically require highly purified, concentrated, monodisperse samples of intact virions with minimal background noise [11].

Common approaches for virus concentration and purification are based on a combination of virus pelleting with ultracentrifugation followed by buffer exchange using dialysis or ultrafiltration [12,13]. Such methods are well suited for icosahedrally-symmetric, sturdy viruses [13,14], but not for pleiomorphic viruses where the particles vary in size, and often lose their integrity during ultracentrifugation [15,16,17,18,19,20,21,22,23,24,25,26]. SARS-CoV-2 is a pleiomorphic virus, producing 50–150 nm diameter particles that consist of a lipid envelope and four structural proteins [27]. The virions typically harbor 25–40 copies of the major glycoprotein—trimeric spike (S)—anchored in the envelope [28]. The correct function of S relies on its structural flexibility, ensured by three hinges within the S stalk [29], multiple conformations of the receptor binding domain [28], and proteolytic processing of S by host proteases like TMPRSS2 and furin [30,31]. Although density gradient-based purification has been proposed for coronaviruses [32], it appears that the S proteins are shed during ultracentrifugation [28]. This shedding occurs even when a stabilized S mutant deficient in furin cleavage is used, and thus this method is suboptimal for producing purified virus for structural studies [28]. As S mediates the host interaction, its presence on the virus is important for vaccine and drug development.

Chromatography is another method used for virus purification. It can efficiently remove contaminants while preserving particle integrity, thus securing good virus recovery [33]. Ion-exchange chromatography on monolithic columns can be used for purification of pleiomorphic viruses. The advantages are that purification and concentration steps are combined, and particle integrity is preserved due to low shearing forces. The disadvantages are that it requires harsh elution conditions and often an additional buffer exchange step [34]. Using size exclusion resins is an alternative that allows purification of pleiomorphic viruses in native buffers [35]. When packed into chromatography columns, the virus may still disintegrate.

Cultures of live SARS-CoV-2 must be handled in biosafety level 3 (BSL3) laboratories, where the procedures are time-consuming and the available equipment is often limited. High-pressure liquid chromatography systems are rarely available in BSL3. Some laboratories are not equipped with ultracentrifuges or do not allow high-speed centrifugation of SARS-CoV-2 due to the aerosol formation risk. Here, using a wild-type SARS-CoV-2 isolate, we present two alternative methods of virus purification using equipment commonly available in BSL3 laboratories. Each method presented can be easily adopted by users to obtain high titer, purified, intact virus preparations for downstream research needs. Furthermore, as SARS-CoV-2 samples may need to be transferred outside of BSL3 containment facilities, we show that UV irradiation, a rapid and efficient infectivity inactivation method which has been previously described for SARS-CoV-2 [36], preserves the structural integrity of the virus particles.

## 2. Materials and Methods

### 2.1. Cells and Lentiviruses

African green monkey kidney epithelial Vero E6 (VE6; ATCC CRL-1586) and human embryonic kidney HEK293T cells were cultured in growth medium (Dulbecco’s Modified Eagle’s Medium (DMEM; Sigma-Aldrich, St. Louis, MO, USA, D6429) supplemented with 10% fetal bovine serum (FBS; Gibco), 2 mM L-glutamax (Gibco), 100 units of penicillin and 0.1 mg/mL streptomycin (PenStrep; Sigma-Aldrich), and 1× non-essential amino acids (NEAA; Sigma-Aldrich). Cells were cultured in a 37 °C incubator with 5% CO_2_ and passaged 1:8 (VE6) or 1:10 (HEK293T) every four days.

To generate non-replicative lentivirus carrying human TMPRSS2 gene (hTMPRSS2), HEK293T cells were seeded in a six-well plate at 1.25 × 10^6^ cells/well and 24 h later were transfected with 825 ng of psPAX2 (gift from Didier Trono; Addgene #12260), 175 ng of pCMV-VSV-g (gift from Bob Weinberg; Addgene #8454) and 1 μg of pLenti6.3-hTMPRSS2 [30] plasmids using TransIT-lenti (Mirus Bio) transfection reagent according to the manufacturer’s instructions. The lentivirus was collected 48 h post transfection, filtered through a 0.45 μm filter and stored at −80 °C until further use.

To generate Vero E6 cells stably expressing hTMPRSS2 (VE6-T), parental VE6 cells were cultured to 60% confluency in growth medium. They were infected with hTMPRSS2-encoding lentivirus at a multiplicity of infection (MOI) of 3 in lentivirus infection medium (DMEM, 2% FBS, 2 mM L-glutamax, PenStrep, NEAA) for 1.5 h, after which the medium was changed to growth medium. At 6 h post infection (h.p.i.) fresh growth medium was added and the cells were incubated until 48 h.p.i. Transductants were selected by maintaining cells in the presence of blasticidin (5 μg/mL) for several passages until all the control non-transduced cells died, after which VE6-T cells were cultured in the growth medium as described above. The expression of hTMPRSS2 was confirmed by immunoblotting using anti-TMPRSS2 antibody (Santa Cruz, CA, USA).

### 2.2. Production of SARS-CoV-2

VE6-T cells were grown to 80–90% confluency in growth medium, washed four times with phosphate-buffered saline (PBS) and the P2 of SARS-CoV2 isolate Finland/1/2020 (Genbank accession number MT020781.2) propagated in VE6-T cell line was added in virus production serum-free medium (VP-SFM, Gibco, 11681020) supplemented with 4 mM L-glutamax (Gibco), at MOI 0.01. Infected cells were incubated at 37 °C and 5% CO_2_. The virus-containing supernatant was collected at 72 h.p.i. and the cellular debris was removed by centrifugation for 5 min at 15 °C, 3220× *g*. A second, MEM-based, infection medium was compared for virus production at 72 h.p.i. (Minimum Essential Medium (MEM, Gibco) supplemented with 0.2% bovine serum albumin (BSA, Sigma Aldrich, St. Louis, MO, USA), L-glutamax, PenStrep and NEAA).

The virus titer was determined using plaque assays. VE6-T cells were seeded on 6-well plates and cultured to 90% confluency in growth medium. Serial 10-fold dilutions of the SARS-CoV-2 stock were prepared in MEM-based infection medium. Cells were washed with PBS and infected with 200 µL of serial virus dilutions. Cells were incubated at 37 °C and 5% CO_2_ for 1 h with rocking, after which the monolayers were covered with 3 mL of overlay medium (MEM, 2% FBS, L-glutamax, Penstrep, 1.2% Avicel (RC-591). After a 60 h incubation, cells were fixed with 4% formaldehyde, washed, and plaques were visualized by staining with a crystal violet solution (0.2% crystal violet, 1% methanol, 20% ethanol, 3.6% formaldehyde). Viral titers were determined as plaque-forming units per mL (PFU/mL) of virus stock.

### 2.3. Sequencing

SARS-CoV2 genomes were isolated from the virus-containing supernatant using a viral RNA extraction kit (Macherey-Nagel, NucleoSpin RNA Virus, 740,956). Viral RNA (3 µL) was transcribed into cDNA using Superscript IV (Thermo Scientific, Waltham, MA, USA) and random hexamers (Thermo Scientific) in a 10 µL reaction as described previously [37]. Multiplex primers for whole viral genome sequencing were designed using Primalscheme [38] with the Wuhan reference sequence (Genbank NC_045512.2) as input. The primers were targeted to amplify ~450 bp overlapping fragments by PCR (Appendix A). Truncated Illumina TruSeq adapter sequences were added to the 5′ end of primers (Appendix A). For the second index-PCR step, Illumina dual-indices were selected using Barcosel [39].

The obtained PCR products were purified using MagSi-NGSPREP Plus magnetic beads (Magtivio) according to the manufacturer’s protocol. The purified PCR products were paired-end sequenced on an Illumina MiSeq Sequencer using a v3 600 cycle kit.

The resulting sequencing paired end reads were trimmed to a minimum read length of 50 bp, minimum Phread quality of 25, and TruSeq adapter removal using Cutadapt v.3.4 [40]. Trimmed reads were mapped with bwa-mem (v.0.7.17) against the SARS-CoV-2 Wuhan (NC_045512.2) reference genome. Primer sequences were removed from the aligned reads using fgbio (v.1.3.0) [41] and the designed primer pair location file. Alignments were sorted and indexed using samtools (v.1.10). Variants were called using bcftools (v1.11) with both mpileup and call analysis.

### 2.4. SARS-CoV-2 Purification/Concentration

Virus stock collected at 72 h.p.i. (Section 2.2) was pre-cleared for 20 min, 10,000× *g* at 15 °C and used for the further purification/concentration methods described below.

#### 2.4.1. SARS-CoV-2 Pelleting

For pelleting, 9.5 mL of pre-cleared virus-containing supernatant was layered on top of a 1.5 mL 20% (*w*/*v*) sucrose cushion in buffer containing 20 mM HEPES and 155 mM NaCl at pH 7.0 (HN) and centrifuged for 2 h at 100,000× *g* at 4 °C. The supernatant was discarded, and the visible pellet was rinsed with HN to remove leftover sucrose and resuspended in 200 µL of HN at 4 °C overnight.

#### 2.4.2. SARS-CoV-2 Purification Using Size-Exclusion Resin

Size exclusion chromatography resin Capto Core 700 (Cytiva, Middlesex County, MA, USA) was washed six times with 10 volumes of HN buffer. A 1.2 mL aliquot of Capto Core (bed volume) was added to 12 mL of pre-cleared supernatant and end-to-end rotated at 4 °C for 20 min. The resin was pelleted by centrifugation for 3 min at 800× *g*, 4 °C and the virus-containing supernatant was collected. The procedure was repeated with a fresh aliquot of Capto Core resin. The purified virus was concentrated to a final volume of 150 µL using 4 mL Amicon Ultra ultrafiltration units with a 100 kD MWCO (Millipore, UFC8100, Burlington, MA, USA).

### 2.5. Mass Spectrometry (MS)

Samples were lysed in lysis buffer (8 M Urea, 50 mM NH_4_HCO_3_ supplemented with 1X phosphatase and protease inhibitors cocktail (Sigma-Aldrich, P2745 and P8340)) on ice. The lysate was cleared by centrifugation at 16,000× *g* for 10 min at 4 °C. The protein concentration was determined using a BCA protein assay kit (Thermo Fisher). Equal amounts of protein (70 µg) were used for all samples. Proteins were reduced with Tris(2-carboxyethyl) phosphine (TCEP; Sigma Aldrich), and alkylated with iodoacetamide. Samples were diluted 4-fold (to less than 2 M urea) with 50 mM ammonium bicarbonate (AMBIC; Sigma-Aldrich), and digested with Sequencing Grade Modified Trypsin (Promega) in an approximate ratio of 1:50 to 1:20 (*w*/*w*) at 37 °C for 16 h. Finally, the peptide samples were desalted with C18 Macrospin columns (Nest Group).

The desalted samples were analyzed using an Evosep One liquid chromatography system coupled to a hybrid trapped ion mobility quadrupole TOF mass spectrometer (Bruker timsTOF Pro) via a CaptiveSpray nano-electrospray ion source. An 8 cm × 150 µm column with 1.5 µm C18 beads (EV1109, Evosep) was used for peptide separation with the 60 samples per day method (21 min gradient time). Mobile phases A and B were 0.1% formic acid in water and 0.1% formic acid in acetonitrile, respectively. The mass spectrometry analysis was performed in positive-ion mode using data-dependent acquisition (DDA) in parallel accumulation-serial fragmentation (PASEF) mode with 10 PASEF scans per topN acquisition cycle.

Raw data (.d) acquired in PASEF mode were processed with MSFragger against the human plus coronavirus entries of the Uniprot database [42,43]. Carbamidomethylation of cysteine residues was used as the static modification. Amino-terminal acetylation and oxidation of methionine were used as the dynamic modification. Trypsin was selected as the enzyme, and a maximum of two missed cleavages were allowed. Both instrument and label-free quantification parameters were left to the default settings.

### 2.6. SDS-PAGE and Western Blot

Samples were mixed with 4× Laemmli sample buffer [44], and proteins were resolved using electrophoresis in a 4–20% sodium dodecyl-sulfate polyacrylamide gel (Mini-PROTEAN TGX, Bio-Rad, Hercules, CA, USA) and visualized using a Gel Doc EZ Imager (Bio-Rad). Proteins were transferred onto a nitrocellulose membrane using a Trans-Blot Turbo transfer system (Bio-Rad). The membrane was blocked using 5% (*w*/*v*) milk in buffer containing 150 mM NaCl, 20 mM Tris and 0.1% TWEEN 20 at pH 7.6 (TBST) for 30 min at room temperature (RT). Membranes were incubated with primary antibodies in 1:1000 dilution in TBST containing 5% milk at RT for 1 h and washed three times for 5 min with TBST. Secondary antibodies were diluted 1:10,000 in TBST, and added on the membranes for 30 min at RT, after which the membranes were washed three times for 5 min with TBST. Proteins were visualized using an Odyssey imager (Li-Cor). The primary antibodies used were rabbit polyclonal anti-SARS-CoV-2-S1, rabbit polyclonal anti-SARS-CoV-2-N [45], and rabbit monoclonal anti-hTMPRSS2 (Santa Cruz). The secondary antibody was IRDye 680 RD goat anti-rabbit IgG diluted 1:10,000 (Li-Cor).

### 2.7. SARS-CoV-2 Inactivation

For inactivation, 50 µL of purified SARS-CoV-2 was exposed to UV-C irradiation at energies ranging from 6.25 to 100 mJ cm^−2^ using a UVP cross-linker CL-1000 (Analytik Jena, Jena, Germany) at A_254_ nm. SARS-CoV-2 samples used for cryoEM were inactivated with 100 mJ cm^−2^. Infectivity reduction was assessed using plaque assays.

### 2.8. Cryogenic Electron Microscopy and Tomography

For cryoEM, virus samples were vitrified on glow-discharged Quantifoil electron microscopy grids with holey carbon film (R1.2/1.3, mesh 300) using a Leica EM GP plunger at 22 °C, 85% humidity and 3 s blotting time using back blotting. For cryoET and particle size distribution analysis of CC purified virus, the sample was concentrated on an ultrafiltration device MWCO 100 KDa (Millipore, Burlington, MA, USA) according to the manufacturer’s instructions. For particle distribution statistics non-concentrated sample was used. For cryoET 5 nm protein A gold was added as a fiducial marker [16]. CryoEM data were collected using an FEI Talos Arctica transmission electron microscope, equipped with a Falcon III direct electron detector at the Helsinki Institute of Life Science CryoEM Core Facility. Data were collected in linear mode at 57,000× nominal magnification at a 2.55 Å/pixel sampling rate using a total dose of 13 e^−^ Å^−2^. To calculate the particle size distribution, 100 particles with a box size of 550 pixels were picked using the manual step of Xmipp3 in Scipion version 2.0 [46,47,48], and classified in to 5 classes using 2D classification in Relion 3 [49,50,51,52]. Classes with the same size average were combined in the calculations. CryoET data were collected on a Titan Krios G2 microscope (Thermo Fisher), equipped with a Gatan BioQuantum energy filter and K3 direct electron detector. A Volta phase plate was used during data collection. Energy-filtered tomographic tilt series were collected with the dose-symmetric scheme [53], as implemented in the Thermo Fisher TEM tomography software. Images were collected at a magnification of 52,000×, corresponding to 2.1 Å/pixel, between −60° to +60°, with 3° steps at −3 µm defocus. The total accumulated dose was approximately 50 e^−^ Å^−2^. Tomographic reconstruction was performed with the IMOD software [54] including CTF-correction for phase-shifted data.

## 3. Results

Vero E6 (VE6) cells are commonly used to produce SARS-CoV-2 as they support virus replication well and do not secrete antiviral signaling mediators to the medium [55]. However, when passaged in VE6, SARS-CoV-2 acquires a deletion upstream of the furin cleavage site (FCS) in S, which affects cell-to-cell spread and tropism [30,56,57]. Such a deletion does not occur in a Vero E6 cell line stably expressing TMPRSS2 (VE6-T) [57]. We therefore generated such a cell line and used it for virus production. We confirmed with sequencing that the progeny virus genome was identical to the parental Finland/1/2020 isolate (Genbank ID MT020781.2).

We compared SARS-CoV-2 production in BSA-supplemented and in serum-free media (VP-SFM), observing similar titers (4 × 10^7^ PFU/mL and 8 × 10^7^ PFU/mL, respectively). The VP-SFM resulted in a lower non-specific protein content due to the lack of BSA. This was assessed by MS giving spectral counts of albumin 556, N 28, S 17, and M 1 in MEM 0.2% BSA medium, whereas VP-SFM spectral counts were albumin 19, N 401, S 187, and M 9. We chose VP-SFM for virus stock production for further experiments.

We produced SARS-CoV-2 in VP-SFM and purified the virus using one (CC1) or two (CC2) rounds of incubation with Capto Core 700 resin, or by pelleting through a 20% sucrose cushion (SP). Virus titers at different steps in the purification showed that compared to the precleared stock (PC) the titer was retained with Capto Core (Figure 1a, Table 1). Pelleting through the sucrose cushion followed by gently dissolving the pellet overnight increased the titer, showing that the sample can be concentrated. We assessed the purity and the presence of viral proteins in the samples using SDS-PAGE and immunoblotting. The SDS-PAGE indicated the presence of the viral N as a major band along with many contaminants in the stock. Capto Core purification efficiently removed most of the contaminants, preserving N, whereas sucrose pelleting concentrated both the virus and virtually all the contaminating proteins (Figure 1b). Immunoblotting indicated a higher concentration of viral proteins in the pelleted sample than in Capto Core, as judged by the detection using anti-S and anti-N antibodies. The presence of S1 indicated that prefusion spikes are still present.

We used MS as a more sensitive method to assess sample purity and virus integrity. We looked at both the total spectral counts of peptides detected from equal amounts of samples, and at the relative amount of SARS-CoV-2 structural proteins normalized to the SARS-CoV-2 non-structural protein Orf9b. Capto core resin removes Orf9b together with other soluble proteins smaller than 700 kDa from the solution. Hence, Orf9b serves as an internal control of the purification. The difference in the total spectral counts indicates that both methods increase purity compared to pre-cleared stock, with two rounds of Capto Core being the most effective (Figure 1c). Comparing relative protein amounts, we see a major increase in the specific viral structural proteins, with a 25-fold increase in N to Orf9b in the case of pelleted virus, and a 35-fold increase when using Capto Core (Figure 1d). The presence of the viral proteins, along with the total yield of infectious virus, indicates that Capto Core was the most efficient at purification (Figure 1d) and retained~100% infectivity (Table 1). In contrast, sucrose pelleting results in a more concentrated virus preparation at the expense of purity.

In order to use purified SARS-CoV-2 outside the BSL3 facility e.g., for cryoEM, virions must be inactivated. UV-inactivation is a fast method that should preserve protein structure [36,58]. We tested inactivation of purified SARS-CoV-2 with increasing energies of UV-C irradiation, observing that the infectivity was reduced from 5 × 10^7^ PFU/mL to non-detectable at 25 mJ cm^−2^ (Figure 2).

Next, we examined the UV-irradiated samples by cryoEM. We collected about 300 micrographs per sample, assessing the number of particles per micrograph and particle size distribution (Table 2 and Figure 3a,b), and comparing them to virus titer (Table 1). We observed over 10-fold higher number of particles per view in the sucrose pelleted samples compared to Capto Core 2× samples. However, the difference in titer of the samples is only 2-fold. Hence, we concluded that many of the particles in the sucrose pelleted sample are non-infectious. The size distribution was tested by 2D classification of 100 random particles from sucrose pelleted and Capto Core 2× concentrated data sets and showed that they were similar (Figure 3c). However, there was extra non-viral material in the sucrose pelleted sample, as expected from the protein analysis. We therefore chose to use electron cryo-tomography of a concentrated Capto Core 2× sample to examine virus morphology. The particles showed typical SARS-CoV-2 morphology with a clear membrane, nucleocapsid and the presence of many pre-fusion spikes on the surface correlating with high virus titer (Figure 3d).

## 4. Discussion

We described a new method for purification of SARS-CoV-2 which retains full infectivity, provides a pure virus sample suitable to use for highly sensitive systems biology methods, and for structural biology. We also confirmed the use of a rapid method for virus inactivation preserving virus morphology. The methods are inexpensive, do not impose an aerosol risk, and do not require specialized equipment.

Recent structural papers of SARS-CoV-2 used chemical fixation and pelleting to concentrate the virus and remove some impurities prior to cryoEM [28,29]. Ke et al. compared the morphology of pelleted and non-purified SARS-CoV-2, concluding that ultracentrifugation of the virus led to spike shearing and was therefore suboptimal for structural work. Other publications on the purification of coronaviruses were based on polyethylenglycol pelleting and ultracentrifugation [22,32]. Our sucrose pelleting experiments agree with these previous studies, providing a concentrated sample, but with suboptimal purity and 97% loss of infection yield. The shearing forces associated with pellet resuspension, for instance, through vigorous pipetting are probably the main culprit causing a loss of infectivity. In contrast, the use of Capto Core slurry retained 100% of infection yield, and based on cryoET, resulted in a good distribution of prefusion spikes on the surface of the virus. The whole protocol can be done in less than 2 h compared to the much longer pelleting-based purification that involved a 2-h centrifugation and a 12-h resuspension. It has been previously reported that Capto Core is a useful medium for purification of other pleiomorphic viruses, including influenza A virus [35]. The advantages of using Capto Core slurry are that it does not require chromatography systems that are often unavailable in BSL3, or ultracentrifugation that is an unwanted source of aerosol. It is fast, scalable, reproducible, suitable for high throughput and GMP adaptable. UV-inactivation is an additional step that can be added to the protocol in case the virus needs to be moved out of contained conditions, e.g., for imaging [36,58]. It is faster than chemical fixation using, e.g., ß-propiolactone or glutaraldehyde, and preserves protein structure allowing the use of the inactivated virus for high resolution structural work.

## Figures and Tables

**Figure 1 viruses-14-01989-f001:**
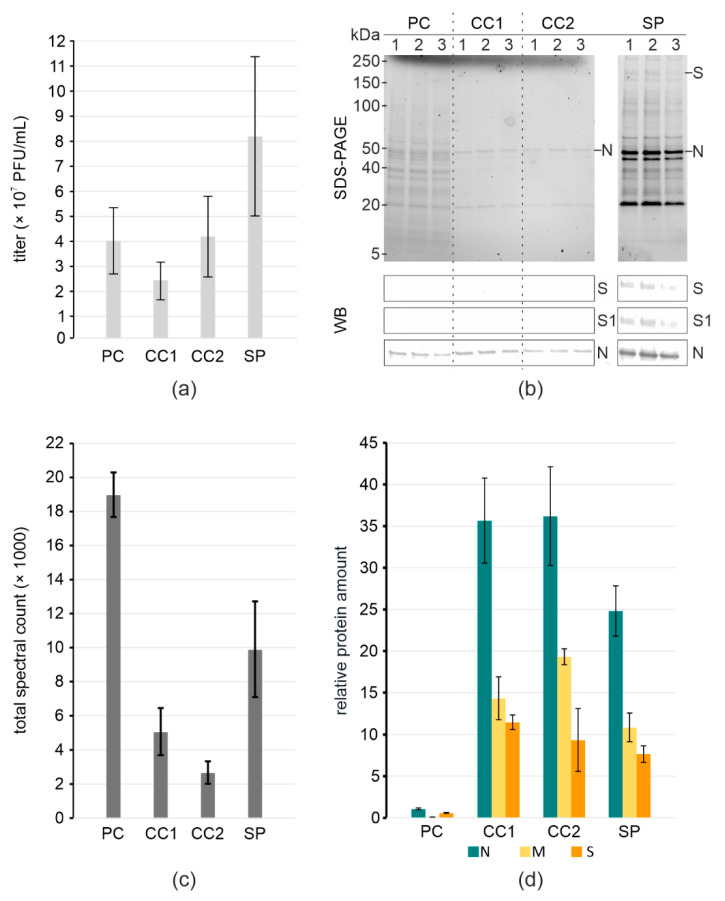
Virus yield and purity at different purification steps. (**a**) Virus titers; (**b)** SDS-PAGE of total protein content and WB of S and N in samples. Molecular weight markers are indicated on the left; (**c**) Total spectral counts of peptides detected in samples; (**d**) Relative protein amount in samples normalized to ORF9b. (**a**–**d**) pre-cleared stock (PC), Capto Core 1× (CC1), Capto Core 2× (CC2); pelleting through 20% sucrose (SP).

**Figure 2 viruses-14-01989-f002:**
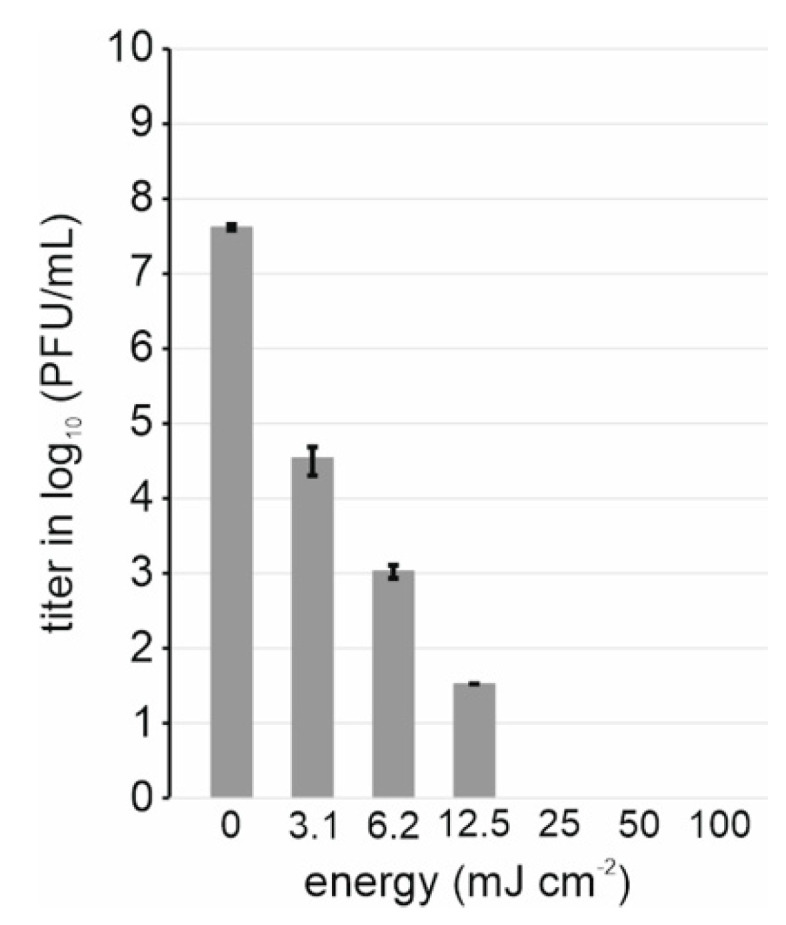
Inactivation of SARS-CoV-2 infectivity by UV. Titers of SARS-CoV-2 after treatment with indicated amounts of UV energy are represented with bars. Bar plot represents data from three independent experiments.

**Figure 3 viruses-14-01989-f003:**
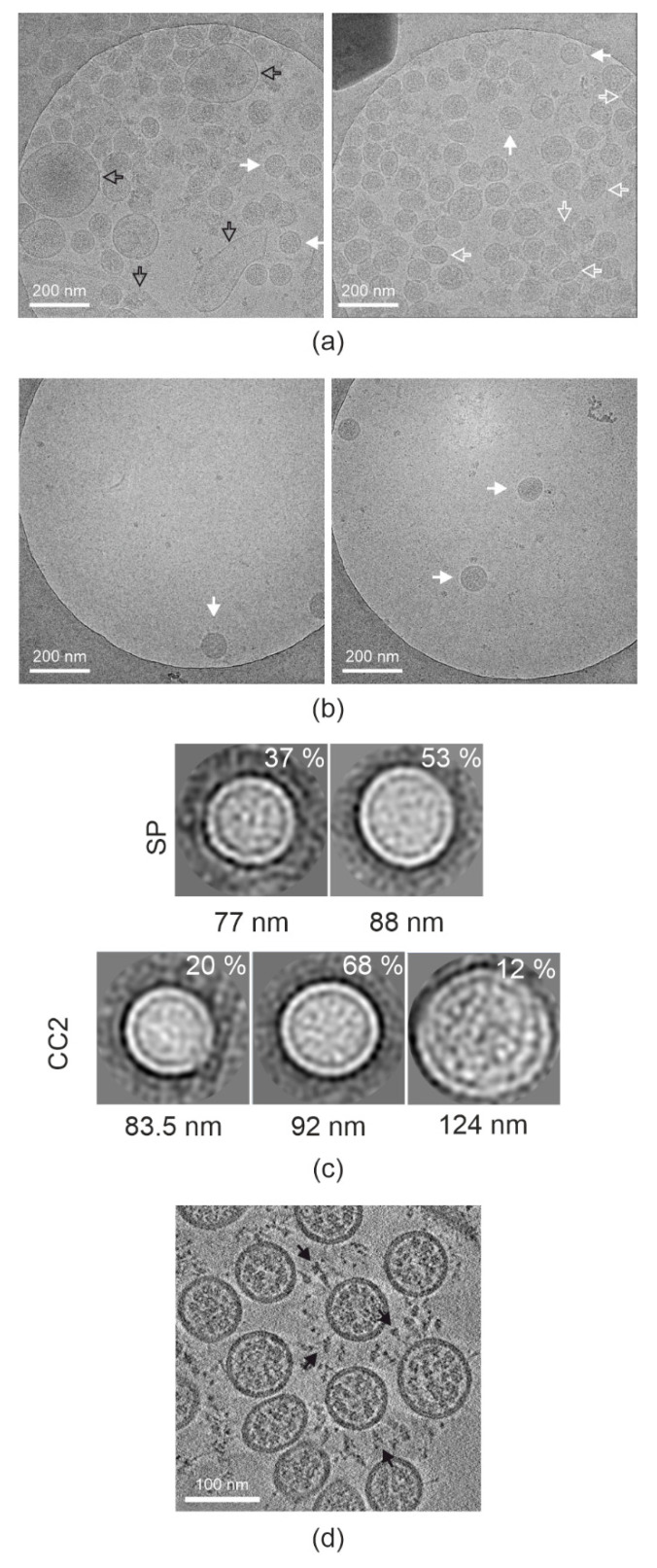
Morphology of inactivated SARS-CoV-2 particles. (**a**) Representative micrographs of SP SARS-CoV-2. (**b**) Representative micrographs of CC2 purified and concentrated SARS-CoV-2. (**a**,**b**) Images taken at −6 µm defocus; Intact SARS-CoV-2 particles (full white arrows), deformed particles (empty white arrows), and contaminating cellular derivatives (empty black arrows) are indicated. (**c**) Class averages of 100 particles picked from the imaged SP micrographs in the top row, and from 100 particles picked from the imaged CC2 concentrated micrographs in the bottom row. The percentage of the particles in the class is shown inside each box, and the average diameter of each class is shown below the box. The box size is 140.25 nm × 140.25 nm. (**d**) A section through a tomogram of concentrated Capto Core 2× purified SARS-CoV-2. Pre-fusion spikes are indicated by full black arrows. Tomogram was collected at −3 µm defocus.

**Table 1 viruses-14-01989-t001:** Comparison of virus yields and purity.

Purification Steps of SARS-CoV-2	Titer (PFU/mL)	Input/Output Volume (mL)	Total Titer (PFU)	Infectious Virus Yield ^a^ (%)
pre-cleared stock	4 × 10^7^ ± 1.3 × 10^7^	12/12	4.8 × 10^8^ ± 1.6 × 10^8^	100 ± 33
Capto Core 1×	2.3 × 10^7^ ± 7.6 × 10^6^	12/12	2.8 × 10^8^ ± 9.1 × 10^7^	58 ± 19
Capto Core 2×	4.2 × 10^7^ ± 1.6 × 10^7^	12/12	5 × 10^8^ ± 1.9 × 10^8^	104 ± 40
Sucrose pelleted	8 × 10^7^ ± 3.1 × 10^7^	12/0.2	1.6 × 10^7^ ± 6.3 × 10^6^	3.3 ± 1.3

^a^ Infectivity retention calculated in comparison to pre-cleared stock.

**Table 2 viruses-14-01989-t002:** CryoEM screening statistics of SARS-CoV-2 samples prepared by different purification methods.

Purification Method	Number of Micrographs	% of Images with Virus ^1^	Total Virus Count ^2^
Capto Core 2x	343	11	40
Sucrose pelleted	281	99	5560

^1^ All images containing viral particles, even if broken, were included. ^2^ Only intact viral particles were considered.

## Data Availability

The mass spectrometry data presented in this study are openly available in the MassIVE database (https://massive.ucsd.edu/) with web access MSV000089770. The SARS-CoV-2 tomogram volume is accessible at DOI 10.5281/zenodo.6787347. All other data presented in this study are available in the article and in the Appendix A.

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
