# Peer review of "SARS-CoV-2 Production, Purification Methods and UV Inactivation for Proteomics and Structural Studies"

_viruses, 2022, doi:10.3390/v14091989_

Round 1

Reviewer 1 Report

This manuscript reports characterization of SARS-CoV-2 virions and evaluates inactivation strategies. Overall speaking, this is an interesting paper. 

Concerns: it is is strange that after CC once, the infectivity drops to 25%. However, the infectivity recovers completely after a second CC. This makes little sense and looks like a random result. Repeats are needed to conclude if this is consistent.

FIgure 3: which populations are the infectious partciles? Do authors plan to study that?

Author Response

We thank the reviewer for the comments. Specifically, we have checked the spelling of the manuscript and as a native English speaker, I have once again gone through to ensure the fluency of the text.

We have redone the plaque assay for the CC1 sample as we only had two replicates for it before, now it is inline with the other samples, and we have updated fig 1a and table 1 accordingly.

The computational analysis leading to two-dimensional class averages of the particles does not allow one to decide which particles are infectious. The presence of prefusion spikes on the surface and nucleocapsid within the particles, indicates particles that at least based on what we know of the infection cycle, have the potential to be infectious. In addition, as we also titered the same preparations, and looked at the recovery of the pfu, most of the particles imaged should be infectious for the Capto Core preparations, but for the sucrose pelleted sample is about 3/100 that are likely to be infectious (Table 2).

Reviewer 2 Report

This manuscript compares two method of virus purification that would be optimal for specific downstream applications.  The manuscript is clearly written, especially the materials and methods as to allow other researchers to complete these techniques.  The manuscript is techniques oriented and is not a hypothesis driven study, potentially providing limited interest or significance.  The title should more specifically describe the intent of the paper. 

Since reference #2 already has a very similar title to this manuscript, a less general title that better reflects the work in this paper would be more appropriate.  Perhaps referring to optimization of protocols to concentrate and purify SARS-CoV-2 for specific downstream purposes. 

Line 39 should specify Vero E6 cells as the virus does not cause CPE in Vero cells.

Lines 60-64 is a very long sentence, should be 2 sentences.  It is unclear what the subject is in line 63.  “and thus (what?) is suboptimal…”

Line 64 “one” should be replaced with “protein”

UV inactivation of SARS-CoV-2 is not new, so should not be described as such.  It has been researched extensively and compared to other inactivation methods.  Some of these references should be included and discussed.

Author Response

We thank the reviewer for the useful comments.

We have made the title longer, to better describe the uses of the purification method. We have tested it for transcriptomics, proteomics and structural studies, but as we only show limited results here for proteomics and structural studies, we only refer to those now in the title. In addition, we specifically state that this is UV inactivation. We did not find structural papers describing the use of UV inactivation for  SARS-CoV-2, although we have used it extensively for other viruses and have also published it for TBEV.  The new title now reads “SARS-CoV-2 production, purification methods and UV inactivation for proteomics and structural studies”

We thank the reviewer for spotting this typo, and it has been corrected to “Vero cell line derivatives”. Case et al. in reference  2 used three types of Vero cell lines, and Vero E6 were what we used.

We have split and rephrased the sentence starting on line 60 to make it clearer, also removing “one” in line 64 in the process. It now reads “Although density gradient-based purification has been proposed for coronaviruses [32], it appears that the S proteins are shed during ultracentrifugation [28]. This shedding occurs even when a stabilized S mutant deficient in furin cleavage is used, and thus this method is suboptimal for producing purified virus for  structural studies [28]. As S mediates the host interaction, its presence on the virus is important for vaccine and drug development.”

We have included the additional reference Patterson et al. (reference 36) to UV inactivation, what we report here is that there is no obvious destruction of the structural integrity of the virus. We also changed the Introduction and Discussion to make it clearer that UV inactivation has been used earlier, but without the structural characterization.